# Optical characterization of surface adlayers and their compositional demixing at the nanoscale

Limin Xiang [1], Michal Wojcik[1], Samuel J. Kenny[1], Rui Yan[1], Seonah Moon[1], Wan Li[1] & Ke Xu [1,2]

Under ambient conditions, the behavior of a solid surface is often dominated by a molecularly thin adsorbed layer (adlayer) of small molecules. Here we develop an optical approach to unveil the nanoscale structure and composition of small-molecule adlayers on glass surfaces through spectrally resolved super-resolution microscopy. By recording the images and emission spectra of millions of individual solvatochromic molecules that turn fluorescent in the adlayer phase, we obtain ~30 nm spatial resolution and achieve concurrent measurement of local polarity. This allows us to establish that the adlayer dimensionality gradually increases through a sequence of 0D (nanodroplets), 1D (nano-lines), and 2D (films) for liquids of increasing polarity. Moreover, we find that in adlayers, a solution of two miscible liquids spontaneously demixes into nanodroplets of different compositions that correlate strongly with droplet size and location. We thus reveal unexpectedly rich structural and compositional behaviors of surface adlayers at the nanoscale.

[1] Department of Chemistry, University of California, Berkeley, CA 94720, USA. [2] Chan Zuckerberg Biohub, San Francisco, CA 94158, USA. Correspondence and requests for materials should be addressed to K.X. (email: xuk@berkeley.edu)

Upon exposure to a liquid or vapor, a solid substrate quickly picks up molecularly thin adlayers that often dominate its surface behavior. For instance, the adsorption of airborne hydrocarbons readily converts an initially hydrophilic metal, semiconductor, or graphene surface into a hydrophobic one within hours[1–4]. Understanding the nature of such adlayers is of utmost importance in addressing the chemistry and physics of both the adsorbate and the substrate, and is thus key to wide-ranging applications from semiconductor fabrication to atmospheric sciences and crude-oil production[3,5–7].

However, it remains a challenge to elucidate the microscopic structure and composition of adlayers. Atomic force microscopy (AFM) and related scanning-probe techniques have been employed to probe adlayer structure[7–10], but are challenging for small-molecule adlayers that only weakly adhere to the surface. Our previous efforts on graphene templating partly overcome this issue, but the influence of graphene-sample interaction is difficult to assess[11–13].

Recent advances in super-resolution fluorescence microscopy (SRM)[14,15], including those based on single-molecule imaging like stochastic optical reconstruction microscopy (STORM)[16] and points accumulation for imaging in nanoscale topography (PAINT)[17], provide noninvasive, optical means to achieve nanoscale spatial resolution. Although originally developed for biology, SRM has proven valuable for non-biological soft-matter and surface systems[18–23]. The recent integration of spectral measurement with SRM further enables multidimensional imaging[24–29]. In particular, with spectrally resolved STORM and PAINT (SR-STORM/SR-PAINT), we have demonstrated four-color SRM[24] and revealed compositional heterogeneity in cell membranes[28].

Here we develop a general approach to probe the nanoscale structure and composition of weakly adhered organic adlayers via spectrally resolved SRM. By recording the images and emission spectra of ~10[6] individual solvatochromic fluorescent molecules that turn fluorescent in the adlayer phase, we optically achieve ~30 nm spatial resolution for the adlayers with concurrent detection of local chemical polarity. Consequently, we reveal how the nanoscale adlayer structure on a glass surface varies as a function of chemical polarity for 8 different solvents, and discover that a solvent mixture spontaneously decomposes into nanodroplets of varying compositions and sizes on the surface.

## Results

**Spectrally resolved super-resolution imaging of organic adlayers.** We employed Nile red, a polarity-sensitive solvatochromic fluorophore[30] that is effectively non-fluorescent in water, but strongly fluorescent in organic phases[31], to achieve SR-PAINT for adlayers of small organic molecules on a glass surface. By illuminating the sample with a total internal reflection (TIR) configuration to excite a depth of ~100 nm from the glass surface, we found that the stochastic insertion of individual Nile red molecules from an aqueous imaging buffer into the adlayers led to bursts of single-molecule fluorescence with low background (Fig. 1a–c). Due to the extreme thinness of the adlayers (a few nanometers; below), single molecules rapidly diffused in (strongly fluorescent) and out (non-fluorescent) of the adlayer. This short timeframe left little chance for lateral diffusion. Consequently, single-molecule fluorescence appeared as diffraction-limited spots with minimal motion blur (Fig. 1b, c).

The resultant single-molecule fluorescence was split for concurrent recording of position and spectrum in the wide field (Fig. 1a)[24,28]. Fig 1b, c each show a small region of a single camera frame (9 ms integration) of the recorded data, for adlayers of trichloroethylene (TCE; Fig. 1b) and chloroform (Fig. 1c), two

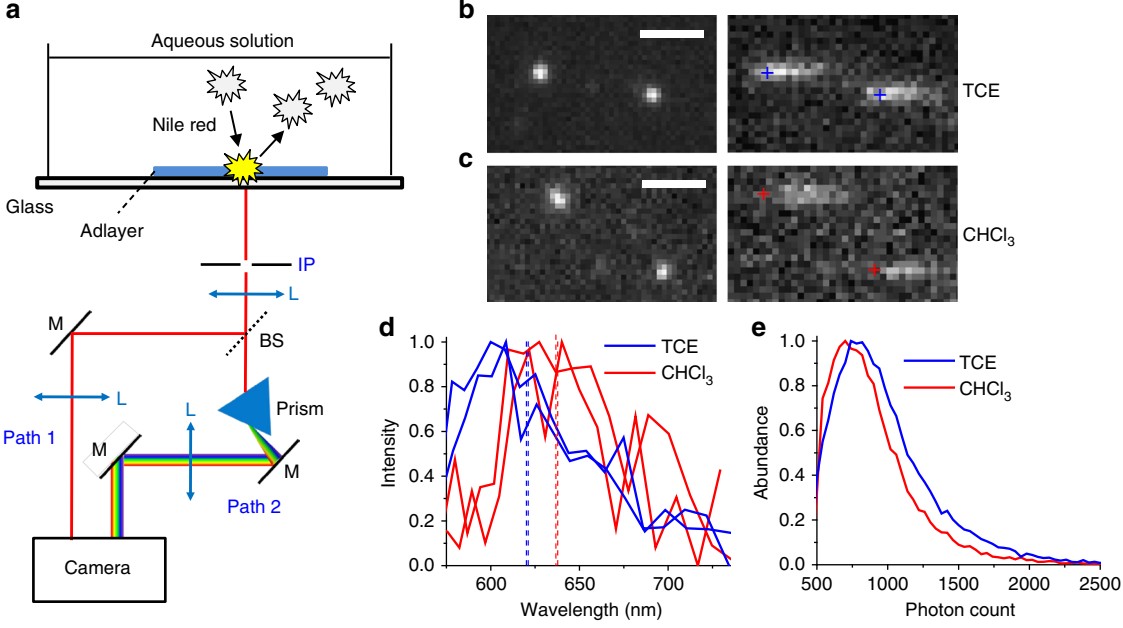

**Fig. 1** Spectrally resolved super-resolution microscopy of small-molecule adlayers. **a** Experimental setup. The stochastic insertion of individual Nile red molecules from an aqueous solution into the TIR-illuminated adlayers leads to bursts of single-molecule fluorescence, which is split into two light paths for concurrent recording of the position (Path 1) and the spectrum (Path 2) of each molecule in the wide field. IP: intermediate image plane, L: lens, BS: beam splitter, M: mirror. **b** A small region of the concurrently acquired images (left) and spectra (right) of two single Nile red molecules inserted into TCE adlayers, obtained in a 9-ms snapshot. Crosses indicate the mapped spectral positions of 590 nm for each molecule. Scale bar is 2 μm. **c** Results of chloroform adlayers. Scale bar is 2 μm. **d** Spectra of the four single molecules in **b**, **c**. Dash lines give the spectral mean of each spectrum. **e** Distribution of photon counts in the image channel for single Nile red molecules in typical data of TCE (blue; average is 984 photons) and chloroform (red; average is 896 photons) adlayers

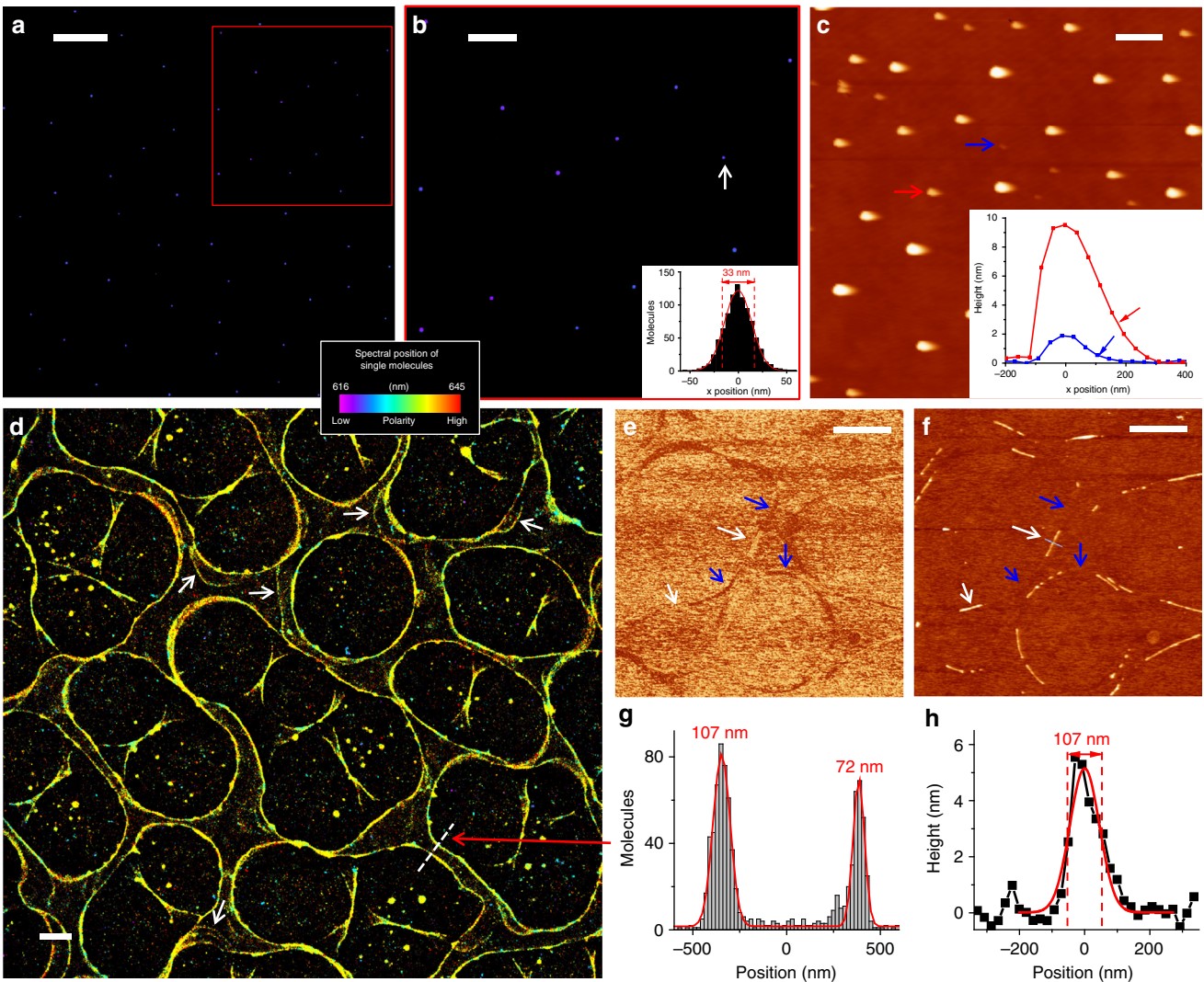

**Fig. 2** Spectrally resolved super-resolution results of TCE and chloroform adlayers. **a** True-color SR-PAINT image of TCE adlayers, color-coded according to the spectral mean values of single Nile red molecules, hence an indicator of local polarity (color bar). Scale bar is 5 μm. **b** Zoom-in of the red box in **a**. Inset: intensity profile for the smallest droplet identified (white arrow). Gaussian fit gives a FWHM width of 33 nm. Scale bar is 2 μm. **c** Soft tapping-mode AFM image of TCE adlayers. Inset: height profiles of two droplets pointed to by the red and blue arrows. Arrows in the profiles point to asymmetry due to dragging. Scale bar is 1 μm. **d** True-color SR-PAINT image of chloroform adlayers, on the same color/polarity scale as **a** and **b**. White arrows point to features between parallel edges. Scale bar is 2 μm. **e**, **f** Soft tapping-mode AFM phase (**e**) and height (**f**) images of the same sample. White and blue arrows point to parts of adlayers that were only visualized in the height and phase images, respectively. Scale bars: 2 μm. **g** Intensity profile along the white dash line in **d** crossing two parallel edges. Gaussian fits give 107 and 72 nm for the FWHM widths of the two edge lines, respectively. **h**, Height profile for the adlayer pointed to by the white arrow at the center of **f**. Gaussian fit gives a FWHM width of 107 nm

common organic solvents of contrasting polarity. Similar single-molecule spectra were observed for the same adlayer, whereas a notable redshift was observed for the more polar chloroform (Fig. 1d), consistent with the solvatochromic behavior of Nile red[31,32]. Statistics of the single-molecule fluorescence intensity in the image channel gave asymmetric single-peak distributions (Fig. 1e) characteristic of typical STORM, PAINT, and other single-molecule experiments[28,33,34], and the averaged counts were ~900 photons. By integrating the recorded positions and spectra of millions of single molecules over different frames, we mapped out both the morphology and polarity of the adlayers at the nanoscale.

**Trichloroethylene and chloroform adlayers**. To present the spectral and spatial information of every detected molecule in one image, we calculated the intensity-weighted average of wavelength for each single-molecule spectrum (dash lines in Fig. 1d), and

used this spectral mean value to assign a color on a continuous scale when plotting the position of each molecule, hence 'true-color' SRM images[24]. To cover the spectral behavior of the different adlayers examined in this work, we mapped a spectral range of 616–645 nm to a violet/blue to orange/red color scale (Figs. 2, 3 and Supplementary Figs. 1-2, 6-11).

In the resultant true-color SR-PAINT SRM images, TCE adlayers consistently appeared as violet-blue colored circular nanodroplets that randomly scattered across the surface (Fig. 2a, b and Supplementary Figs. 1-2). The nanodroplets were often <~100 nm in diameter (Fig. 2a, b), significantly smaller than the optical diffraction limit (~300 nm), although larger ones up to several hundred nanometers were also observed (Supplementary Fig. 1). The smallest nanodroplets had apparent sizes of ~33 nm in full width at half maximum (FWHM) (Fig. 2b inset), limited by our image resolution. While the ~900 photons we detected per molecule (Fig. 1e) could theoretically translate to a single-

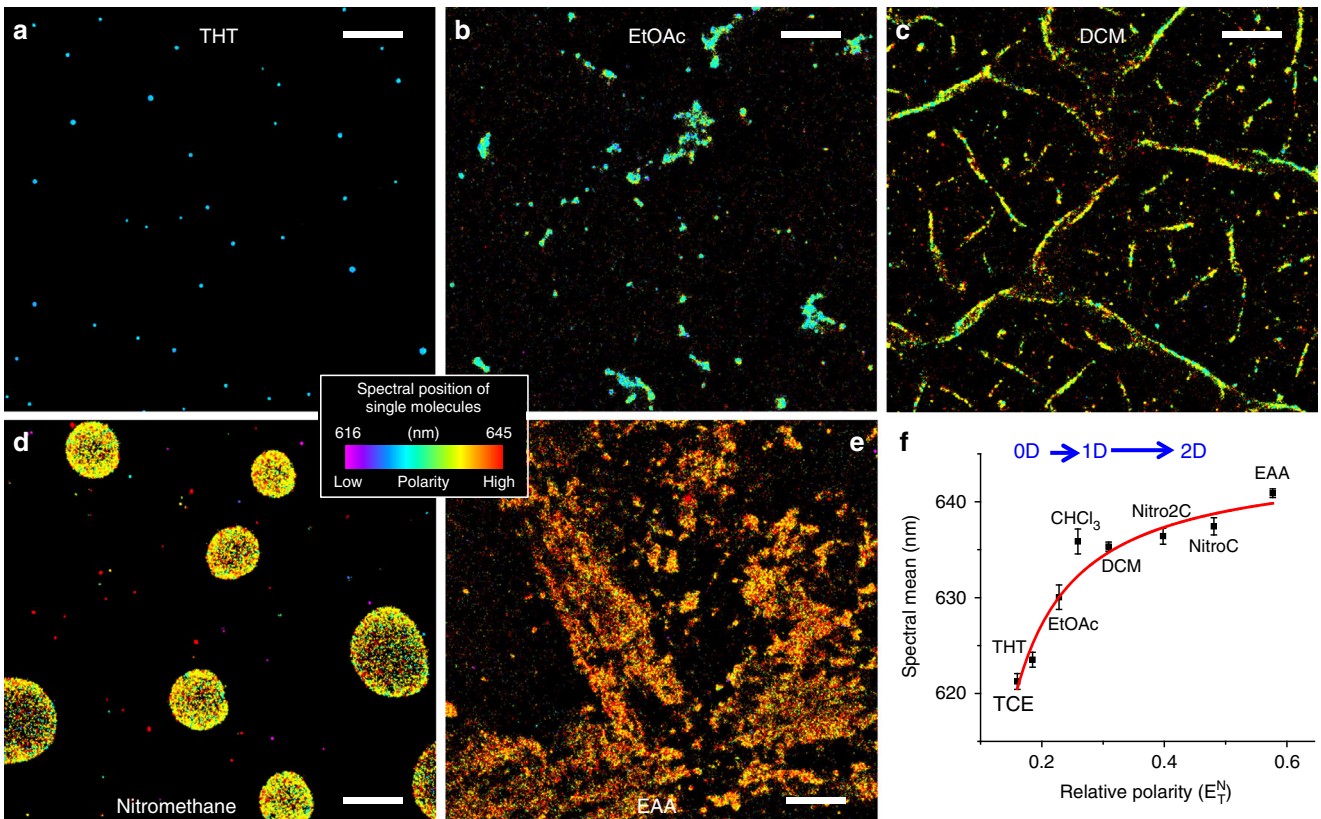

**Fig. 3** Dependence of adlayer spectral and structural properties on polarity of the liquid. **a–e** True-color SR-PAINT images of adlayers of tetrahydrothiophene (**a**) ethyl acetate (**b**) dichloromethane (**c**) nitromethane (**d**) and EAA (**e**). Images were color-coded according to the spectral mean values of single Nile red molecules, on the same spectral/polarity scale as Fig. 2. Scale bars: 2 μm. **f** Measured spectral mean values of the adlayers vs. the relative polarity (normalized Dimroth–Reichardt ET scale[30]) of the liquids. THT: tetrahydrothiophene, EtOAc: ethyl acetate, DCM: dichloromethane, Nitro2C: nitroethane, NitroC: nitromethane. Each data point is averaged from 9–12 areas of 3–6 different samples, with standard deviations drawn as error bars. Red curve is a guide to the eyes

molecule localization precision of ~17 nm in FWHM[35,36], experimentally a ~30 nm FWHM localization precision has been found under similar settings[28,37] due to imperfections of the imaging system including pixel nonuniformity and mechanical instability[38]. Deconvolution of this precision/resolution from our SR-PAINT data implies that the true sizes of these smallest nanodroplets were <20 nm.

Markedly different spectral and structural characteristics were observed in the SR-PAINT data of the more polar chloroform. Adlayers appeared yellow on the same spectral scale, indicating higher polarity, and were self-organized into circular networks ~5 μm in diameter across the substrate (Fig. 2d and Supplementary Figs. 1-2). Edges of the networks were ~100 nm in width, again significantly below the diffraction limit of conventional optical methods, and often appeared as parallel pairs at ~1 μm separation (Fig. 2g), suggesting a mechanism in which initially ~1 μm wide adlayer strips gradually dewet into ~100 nm-wide lines along their two edges. Accordingly, additional residual adlayers were occasionally observed between the parallel edges (white arrows in Fig. 2d).

We note that the adlayers were stable with the application of the imaging buffer necessary for our SR-PAINT approach. Among the different adlayers investigated in this work, we found that the TCE droplets could be directly observed by differential interference contrast (DIC) microscopy, albeit at much reduced resolution (~600 nm) (Supplementary Fig. 2). Adding the imaging buffer did not alter the adlayer structure (Supplementary

Fig. 2). Through SR-PAINT, we further confirmed the adlayer nanostructures of both TCE and chloroform to be stable in the imaging buffer over hours, in terms of both nanoscale structure and polarity (Supplementary Fig. 2). The observed high structural stability may be due to the known enhanced adlayer-substrate interactions at the nanoscale, as manifested by drastically reduced contact angles found in previous studies[13,39] and below.

We have compared PAINT results obtained with another dye, Merocyanine 540, which is also characterized by a substantial increase in fluorescence in the organic phases when compared to the aqueous phase[40]. Comparable adlayer structures were observed (Supplementary Fig. 3). However, when compared to Nile red, Merocyanine 540 exhibits minimal shifts in emission spectrum for solvents of different polarities[40] (Supplementary Fig. 3), and so is not useful for revealing local polarity. Nonetheless, the consistency of results from different dyes indicates that the choice of fluorescent probe has no impact on the image generation process.

We next compared results with AFM acquired in the dry state in air. Soft tapping-mode using a probe of low (~5 N m$^{-1}$) force constant visualized structures consistent with our SRM results, but strong disturbances to the adlayers were apparent. Specifically, for the TCE adlayers, asymmetric droplet shapes were observed along the scanning direction (Fig. 2c), indicative of dragging by the scanning tip. Such artifacts from tip-adlayer interactions[8] became more severe when scanning at high magnifications, and so the shape of the smaller (<~50 nm)

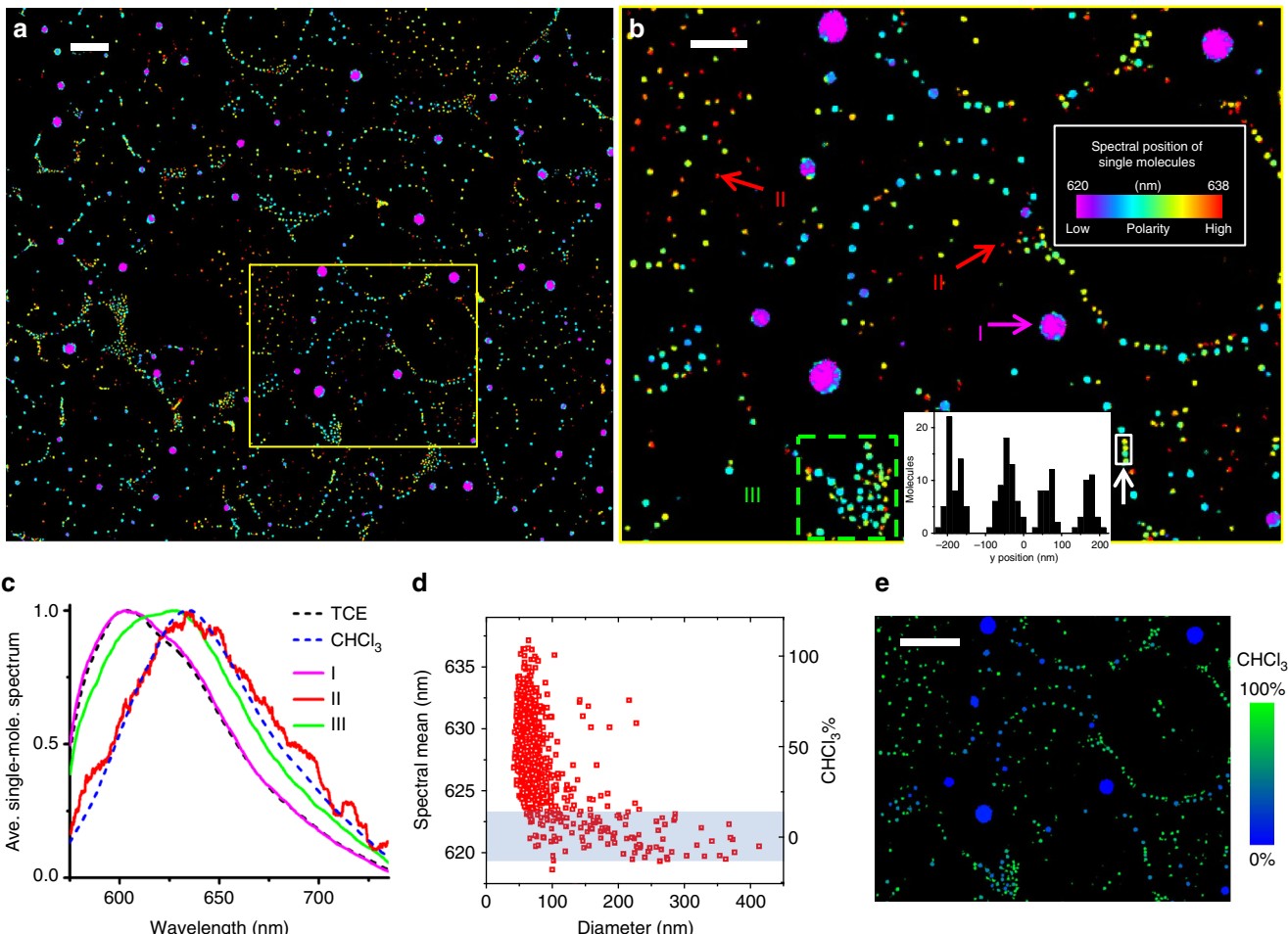

**Fig. 4** Nanoscale decomposition of a two-component mixture on the surface. **a** True-color SR-PAINT image of adlayers from a 1:3 mixture of TCE and chloroform, drawn on a spectral scale of 620–638 nm. Scale bar is 2 μm. **b** Zoom-in of the yellow-boxed area in **a**. Inset: intensity profile along the vertical (*y*) direction for the four closely located nanodroplets marked by the white box. Scale bar is 1 μm. **c** Averaged single-molecule spectra obtained for Regions I (large nanodroplets), II (the reddest small nanodroplets), and III (small nanodroplets with varying color) in **b** (solid lines), compared to that of adlayers of pure TCE and chloroform (dash lines). **d** Averaged spectral mean value (left *y*-axis) and corresponding chloroform percentage (right *y*-axis) of each droplet, as a function of droplet size. Shaded band represents the typical range of nanodroplets of pure TCE adlayers. **e** Recoloring of **b** by the estimated chloroform percentage of each nanodroplet. Scale bar is 2 μm

droplets was difficult to determine (Supplementary Fig. 4). Typical heights of the nanodroplets were a few nanometers (Fig. 2c inset), thus indicating very-flat geometries with contact angles of <~10°, consistent with the known behavior of adsorbed nanodroplets[13,39]. For the chloroform adlayers, AFM showed fragmented adlayer structures so that the circular networks were, intriguingly, only partly visualized in either the phase (Fig. 2e) or the height (Fig. 2f) images in a complementary fashion (blue and white arrows), again signifying problems due to tip-adlayer interactions. For regions where height data appeared normal, the network edges were ~100 nm in width (Fig. 2h), consistent with SR-PAINT results (Fig. 2g), and a few nanometers in height, again indicating flat geometries. Tapping mode with a standard probe (~48 N m$^{-1}$) gave reduced adlayer contrast, whereas in contact mode, the AFM tip severely dragged the adlayers along, and so no adlayers were visualized for either TCE or chloroform (Supplementary Fig. 5).

**Dependence of adlayer spectral and structural properties on solvent polarity.** The contrasting spectral and structural

characteristics of TCE and chloroform adlayers revealed by SR-PAINT prompted us to investigate whether universal structural and polarity trends can be established. We thus next examined adlayers of 6 more common solvents, namely tetra-hydrothiophene, ethyl acetate, dichloromethane, nitroethane, nitromethane, and ethyl acetoacetate (EAA) (Fig. 3a–e and Supplementary Figs. 6-11). Together with the above-discussed data of TCE and chloroform (Fig. 2a, b, d and Supplementary Figs. 1-2), we found that the single-molecule spectra of Nile red at different adlayers were largely uniform within adlayers of the same liquid, but redshifted for higher solvent polarity. Plotting the averaged spectral means for single molecules at the different adlayers as a function of the relative polarity of the liquids showed a near-monotonic trend (Fig. 3f). These results indicate that the polarity of the adlayers is well correlated with that of the bulk liquid.

Remarkable polarity-dependent trends were also found for adlayer morphology at the nanoscale. The two low-polarity liquids, TCE and tetrahydrothiophene, both showed up as zero-dimensional (0D) nanodroplets across the surface (Figs. 2a, b, 3a, Supplementary Figs. 1-2 and 6). With increased polarity, adlayers of chloroform and dichloromethane both appeared as intertwined

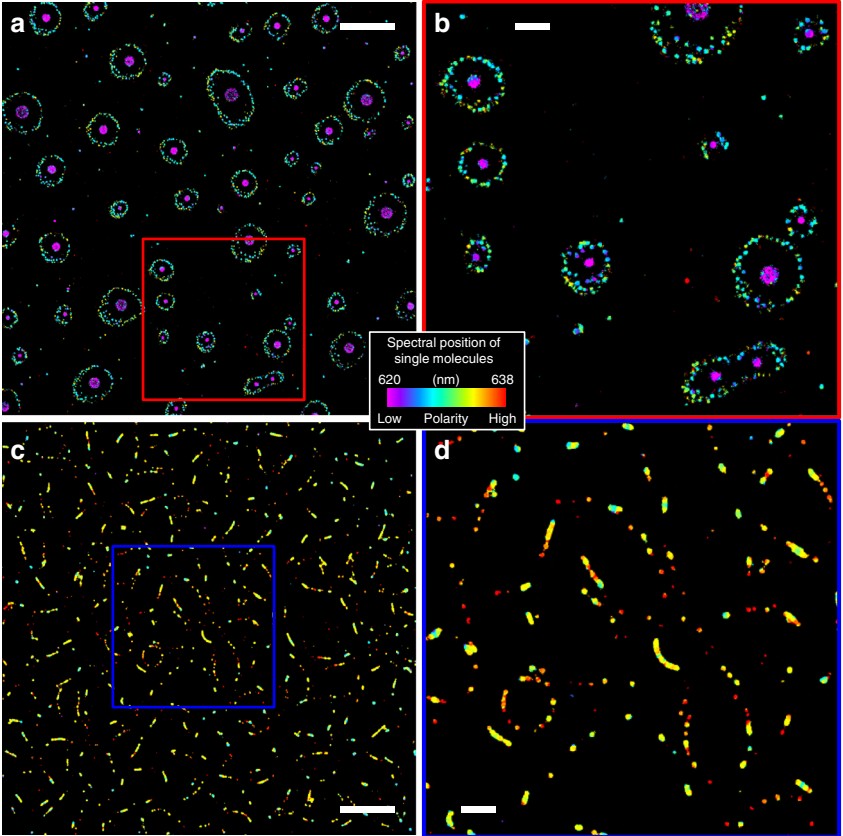

**Fig. 5** SR-PAINT reveals the effect of mixture composition on adlayer structure and composition for the TCE-chloroform system. **a**, **b** True-color SR-PAINT images of adlayers from a 1:1 TCE-chloroform mixture, on the same spectral scale as Fig. 4 (620–638 nm). **b** is a zoom-in of the red box in **a**. **c**–**d** True-color SR-PAINT image of adlayers from a 1:6 TCE-chloroform mixture. **d** is a zoom-in of the blue box in **c**. Scale bars: 4 μm (**a**, **c**) and 1 μm (**b**, **d**)

networks of one-dimensional (1D) nano-lines (Figs. 2d, 3c, Supplementary Figs. 1, 2, and 8). With polarity in between the above two cases, ethyl acetate gave nanoclusters of in-between morphology (Fig. 3b and Supplementary Fig. 7). Liquids of relatively high polarity—namely, nitroethane, nitromethane, and EAA, spread well to form large domains of two-dimensional (2D) films micrometers to tens of micrometers in size (Fig. 3d, e and Supplementary Figs. 9-11). Our results hence establish a model in which adlayer dimensionality at the nanoscale gradually increases with polarity on the hydrophilic glass surface.

We again compared AFM results. Although soft tapping-mode AFM performed adequately for ethyl acetate (Supplementary Fig. 7), it did not visualize any dichloromethane adlayers (Supplementary Fig. 8). This result may be attributed to the very-low boiling point of dichloromethane (39.6 °C); the adlayers may thus be too fragile to be probed by an AFM tip. Meanwhile, nitromethane adlayers were barely visible in AFM due to their very-small height (~1 nm) and tip-adlayer interactions (Supplementary Fig. 10). Overall, our AFM results, acquired in the dry state in air, were generally consistent with our PAINT SRM results, although tip-sample interactions often obscured nanoscale structural features visualized by PAINT.

**Spontaneous nanoscale demixing of a two-component mixture**. The distinct spectra we resolved in SR-PAINT for adlayers of different liquids suggest a mechanism to distinguish nanoscale adlayers of unknown compositions, e.g., those formed from mixtures. Remarkably, by immersing a glass surface in a 1:3 well-mixed solution of TCE and chloroform, two liquids miscible in

bulk, the resultant adlayers showed up as a palette of different colors and nanoscale structures in SR-PAINT images (Fig. 4a, b and Supplementary Fig. 12; note that colors are mapped to a narrower spectral range of 620–638 nm). Overall, adlayers appeared as both isolated, large (dia. ~200 nm) nanodroplets of bluer spectra, and smaller (<100 nm), redder droplets that were arranged into fragmented networks. Both the sizes of the droplets and the separations between the droplets were often substantially smaller than the diffraction-limited resolution of conventional light microscopy (Fig. 4b inset). The large nanodroplets had averaged single-molecule spectra that matched well to adlayers of pure TCE (Fig. 4c). In the other limit, the reddest small nano-droplets exhibited single-molecule spectra that agreed with that of adlayers of pure chloroform (Fig. 4c). Meanwhile, a majority of the small nanodroplets showed intermediate spectra (Fig. 4c), suggesting varying compositions. These results indicate that although we started with a well-mixed solution, TCE and chloroform spontaneously demixed on the glass surface to form adlayer nanodroplets of different compositions. Although the nanoscale demixing of miscible liquids has been reported in confined systems[41,42], related effects for surfaces are usually dis-cussed within the context of the dewetting of films[43–45]. The unique capability to resolve local compositions allowed us to reveal nanoscale demixing in surface adlayers.

To quantify the composition of every adlayer nanodroplet, we calculated the averaged spectral mean value for all detected single molecules within each nanodroplet. Plotting this result as a function of the size of the nanodroplets (Fig. 4d), both of which were uniquely delivered by SR-PAINT, indicated that most

droplets >~100 nm in diameter had averaged spectra similar to pure TCE adlayers. For the smaller droplets, a very limited fraction attained spectra comparable to that of adlayers of pure chloroform, whereas the others showed a continuous distribution of wavelengths between the two limits. Using a simple linear interpolation (secondary y-axis of Fig. 4d), we directly converted the averaged spectral position of each nanodroplet to its mole fraction of chloroform, thus facilitating simultaneous visualization of the nanodroplet composition, size, and location (Fig. 4e).

The scattered distribution of large nanodroplets of ~100% TCE is similar to results on adlayers due to pure TCE (Fig. 2a, b). Meanwhile, the smaller nanodroplets, with varying fractions of chloroform, were arranged into ring-like network patterns that are reminiscent of adlayers of pure chloroform (Fig. 2d), albeit here the networks were broken into nanodroplets. These intriguing results, wherein adlayer composition correlates strongly with the droplet size and location, point to an intricate local balance and/or competition between the vapor pressures and surface interactions of TCE and chloroform at the nanoscale.

We further examined the influence of mixture composition by varying the ratio of TCE and chloroform in the starting solution (Fig. 5). For adlayers of a 1:1 TCE-chloroform mixture (Fig. 5a, b), fewer high-polarity droplets were observed when compared to the adlayers of the 1:3 TCE-chloroform mixture (Fig. 4), and small, intermediate-polarity droplets surrounded large low-polarity droplets as opposed to forming fragmented networks. This result is consistent with the lower chloroform concentration in the starting mixture: the vapor pressure of chloroform is 3-fold higher than TCE at room temperature, so it evaporates away faster during adlayer formation. In contrast, for adlayers of a 1:6 TCE-chloroform mixture (Fig. 5c, d), the low-polarity large droplets disappeared, and segments of nano-lines interspersed with small droplets, consistent with the higher chloroform content of the starting mixture. Together, our results on adlayers of different starting mixtures showed trends consistent with the expected physical properties of the two solvents, but uniquely revealed the remarkable evolution of nanoscale structures and composition distributions of this system.

## Discussion

By recording the fluorescence spectra and positions of millions of single solvatochromic molecules that turn fluorescent in the organic phase, our SR-PAINT approach uniquely allowed for the nanoscale visualization of the morphology and composition of weakly bound adlayers of small organic molecules. Through examination of 8 different molecules, we first established general trends for how adlayer geometry/dimensionality evolves as a function of molecular polarity. Although corresponding AFM results can be interpreted as consistent with our SR-PAINT data, tip-adlayer interactions often disturbed the sample and led to incomplete visualization of the adlayers. The morphology information alone also did not lend confidence to whether all observed structures were due to adlayers. In contrast, with SR-PAINT, we detected uniform single-molecule spectra for adlayers of the same solvent, as well as consistent spectral redshifts for the more polar solvents, thus confirming the identity of the observed structures while also showing that the adlayer polarity is well correlated with that of bulk solvent.

Taking this unique spectral information to the next level, we examined adlayers from a solvent mixture, and revealed that for the miscible TCE-chloroform system, in adlayers the two solvents spontaneously demix into nanodroplets of varying compositions. Notably, the composition correlated strongly with droplet size

and location, thus pointing to rich interactions between different factors in determining how adlayers are formed from solvent mixtures. While such capabilities are immediately powerful for probing adlayers due to different chemical and physical processes (e.g., our preliminary results on the influence of mixture composition on adlayer morphology and composition; Fig. 5), our spectrally resolved SRM approach also opens the door to the nanoscale structural and functional interrogation of other similarly challenging surface and soft-matter systems inaccessible to current methods.

## Methods

**Materials and sample preparation.** Trichloroethylene (TCE) (99%), tetra-hydrothiophene (98%), nitroethane (98%), nitromethane (98%), ethyl acetoacetate (EAA) (99%), and glycine (99.5%) were from Alfa Aesar. Chloroform (99.8%) was from BDH Chemicals. Dichloromethane (99.5%) was from Sigma-Aldrich. Ethyl acetate (99.9%) was from Fisher Chemical. Nile red (99%, Acros Organics) was dissolved in dimethyl sulfoxide to form a 3 mM stock solution and kept at $-20\,°C$. Glass coverslips (25 mm circle, VWR) were cleaned with a heated piranha solution (75% sulfuric acid and 25% hydrogen peroxide), and then rinse with Milli-Q water (18.4 $M\Omega$ cm), thus rendering a highly hydrophilic surface. After blown dry with nitrogen, the coverslip was immersed into a chosen liquid for 3 hr, and then let dry in air.

**Optical setup.** SR-PAINT SRM experiments were performed on a home-built setup (Fig. 1a) based on a Nikon Ti-E inverted fluorescence microscope. A 561-nm laser (Coherent) was introduced onto the back focal plane of an oil-immersion objective lens (Nikon CFI Plan Apochromat λ 100×, NA 1.45) via a dichroic mirror (ZT561rdc, Chroma). A translation stage shifted the laser beam toward the edge of the objective so that emerging light reached the sample at an incidence angle close to the critical angle of the glass–water interface to achieve TIR wide-field illumination. Emission was filtered by a long-pass (ET575lp, Chroma) and a short-pass (FF01-758/SP, Semrock) filter, and then cropped at the image plane of the microscope camera port to a width of ~4 mm. The cropped intermediate wide-field image was collimated by an achromatic lens ($f = 80$ mm) and then split into two perpendicular paths with a 50:50 beam splitter (BSW10, Thorlabs). In Path 1, emission was focused by an achromatic lens ($f = 75$ mm) onto one-half of an EM-CCD (electron-multiplying charge-coupled device) camera (iXon Ultra 897, Andor) to achieve an effective magnification of ~94× for wide-field recording of single-molecule images. In Path 2, emission was dispersed by an equilateral calcium fluoride prism (PS863, Thorlabs) and then focused by an achromatic lens ($f = 60$ mm) onto the other half of the EM-CCD to generate spectra of the same single molecules in the wide field. Wavelength calibration was performed by using fluorescent beads and narrow bandpass filters[24]. Briefly, 100-nm diameter, four-color fluorescent beads (T7279, Life Technologies) were adsorbed to a glass coverslip at low density, and imaged on the setup with 405 nm, 560 nm, or 647 nm excitation. Beads appeared as diffraction-limited spots in Path 1, and as dispersed 1D spectra in Path 2. Bandpass filters with ~10 nm bandwidth were used to determine the spectral positions of different known wavelengths in Path 2 relative to the bead positions in Path 1.

**SR-PAINT imaging of adlayers.** The coverslip sample was immersed in a 10 mM glycine buffer solution (pH = 9.4) containing ~3 nM Nile red, and was continuously illuminated with the 561 nm laser under the abovementioned TIR configuration at an intensity of ~2 kW cm$^{-2}$. The stochastic insertion of individual Nile red molecules from the imaging buffer into the adlayers led to transient bursts of single-molecule fluorescence, whereas the background signal was low due to the very-low quantum yield of Nile red in water and the TIR illumination. Single-molecule fluorescence bursts usually switched off within the same camera frame they appeared, as the Nile red molecules returned to the aqueous phase or were photobleached. Single-molecule fluorescence emission was concurrently recorded in the wide-field as non-dispersed images (Paths 1) and dispersed spectra (Path 2) (Fig. 1a–c). Sparsity of single-molecule fluorescence in the imaging buffer was achieved by adjusting Nile red concentration in the imaging buffer to avoid signal overlapping between molecules. The EM-CCD recorded continuously at 110 frames per second (integration time: 9 ms per frame) for a frame size of 512 × 256 pixels (256 × 256 pixels for wide-field single-molecule images (Path 1) and spectra (Path 2), respectively), and typically collected 20,000−40,000 frames (3–6 min) for each experiment. Approximately 900 photons were collected for each molecule in the image channel (Fig. 1e), and the typical signal-to-noise ratio is >25 for each molecule. To correlate the spatial and spectral positions of Path 1 and Path 2, a narrow bandpass filter centered at 590 nm (FF01-590/10, Semrock) was placed before the beam splitter for ~5,000 frames. Correlation of the single-molecule images in Path 1 and the 590 nm-filtered single-molecule spectral images in Path 2 generated a mapping function between the two channels. Using this mapping function, the super-

localized positions of single molecules in Path 1 were projected to the coordinates of Path 2. Single-molecule spectra were obtained through the mapped position of 590 nm and the aforementioned calibration curve. The intensity-weighted average of wavelengths for each single-molecule spectrum was taken as the spectral mean. This value was used to assign the color to the single molecule based on a continuous color scale, hence 'true-color' super-resolution images.

**AFM characterization**. Unless otherwise noted, AFM images were taken in the dry state in air on an Asylum MFP-3D system in soft tapping mode using aluminum-coated probes (Tap150Al-G; BudgetSensors). Nominal values of the force constant, resonance frequency, and tip radius were 5 N m$^{-1}$, 150 kHz, and <10 nm, respectively. No attempts were made to deconvolve the tip geometry in the presented data. Supplementary Fig. 5d–f were taken in contact mode using the same probes. Supplementary Fig. 5a–c were taken under standard tapping mode using a probe with a nominal force constant of 48 N m$^{-1}$ (PPP-NCL-50; NANOSENSORS).

**Data availability**. The datasets generated during and/or analyzed during the current study are available from the corresponding author on reasonable request.

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

## Acknowledgements

We thank Meghan Hauser for help in the construction of the optical setup. This work was supported by the Beckman Young Investigator Program, the National Science Foundation (CHE-1554717), and the Packard Fellowships for Science and Engineering. K.X. is a Chan Zuckerberg Biohub investigator and acknowledges support from the Bakar Fellows Award. M.W. acknowledges NSF Graduate Research Fellowship (DGE-1106400). S.M. acknowledges a Samsung Scholarship.

## Author contributions

K.X. conceived the idea. L.X. performed the optical experiments and analyzed the data. M.W. performed AFM characterization. S.J.K. and W.L. wrote the data analysis program. S.M., R.Y., and K.X. helped with the data analysis. All authors wrote the manuscript.

## Additional information

**Competing interests:** The authors declare no competing interests.

