## [Peer Review File · Nature Communications]

Reviewers' comments:

Reviewer #1 (Remarks to the Author):

1.) The localization position shown in the figure 2b can originate from the localization precision of a single molecule and not necessarily has to be limited by the confinement of the droplet.

A more detailed explanation by the authors would be required. In particular because there AFM data on small droplets (suppl. 1) is not convincing (tip artefacts).

2.) The authors conclusion in the figure 2b, assuming the size of the smallest being limited by the localization precision, may have a different reason. The localized fluorescent signal may also be a response function of an individual molecule, which is immobilized/diffusing in a larger/smaller nanodroplet. Can the number of molecules in a nano-droplet be estimated? The correlative AFM does not prove the suggestion. A single molecule intensity analysis may give additional information on this issue. Did the authors observed diffusion of the molecules in the adlayers? All the mentioned information will be required to prove the statement. Can the authors improve their argumentation, or add new, more detailed experimental data?

3.) Most of the droplets and structures shown in the manuscript have a substructure sizes > than the optical diffraction limit. What is the advantage of PAINT in this particular case?

Minor remarks:

4.) The AFM images showing the smallest droplets are distorted by the AFM-scan. Did authors try to use different cantilevers, having tips made of various materials?

5.) What is the SNR for the fluorescent Nile Red signals? What illumination time? How much time is required for full sample characterization? How much faster is the characterization of a single diffraction limited image?

6.) What are the time points of measuring? Are the observed structure stable in time?

7.) Axis labeling in figures is not consistent (e.g. brackets in Fig. 2 g and h, 3f and 4c in comparison to the physically correct division by the unit as shown in Fig. 1d)

8.) Color scale bars are missing regarding the height in AFM images.

Reviewer #2 (Remarks to the Author):

Point-wise Comments:

- o This paper deals with study of structure and composition of adlayers of small molecules of differing polarity on glass surface using the novel spectrally resolved super-resolution microscopy technique that the authors have reported earlier [Zhang, Z. Y., Kenny, S. J., Hauser, M., Li, W. & Xu, K. Ultrahigh-throughput single-molecule spectroscopy and spectrally resolved super-resolution microscopy. Nat. Methods 12, 935-938 (2015)].
- o Authors have established the dimensionality and polarity of the different adlayer molecules on the glass surface by imaging and collecting the emission spectra of polarity-sensitive solvatochromic fluorophore Nile Red molecule stochastically inserted into adlayers of various small organic molecules by employing spectrally resolved points accumulation for imaging in nanoscale topography (SR-PAINT) technique along with tapping mode-AFM for height determination of the adlayers.
- o They also report interesting and direct visualization of the demixing at nanoscale for adlayers formed from a solution of two miscible solvents.
- o The mapping of the changing structural details and their correlation with the polarity of the different organic molecules by a non-invasive optical SR-PAINT technique and its confirmation with AFM topography is novel concept.
- o This technique may provide very valuable information in the field of self-assembled monolayers and their applications.
- o In the introduction when referring to Scanning probe techniques having been used to probe adlayer structure, a reference to the role of ambient water adlayers formed on graphene interfaces on hydrophilic and hydrophobic substrates as reported by Gargi Raina et al. [Phys.Chem.Chem.Phys., 2015, 17, 13964] is appropriate.
- o The ability to reproduce the work by other researchers is fair provided the experimental set-up can be assembled appropriately.

Points of Concern:

- o Though this optical technique would help in visualizing the structure of small molecule adlayers on various technologically important substrates non-invasively, the role of the chosen fluorophore molecule's effect on the image generation process needs to be understood more clearly.
- o The demixing at nanoscale for adlayers formed from a solution of one particular ratio of 1:3 of TCE and Chloroform miscible liquids shows the formation of nanodroplets of different sizes and

varying shapes. It would be interesting to have a detailed study of various compositions of the two miscible liquids and their effect on the size and arrangement of adlayer molecules on the surface.

In conclusion, this manuscript can be accepted as it is on account of the demonstrated novel application of SR-SRM tool for the study of the structure of the adlayer composition of small molecules on surfaces and thus providing useful and important insights into the surface structure dynamics of the different types of adlayers in general, at the nanoscale.

Reviewer #3 (Remarks to the Author):

This paper reports an original application of super-resolution fluorescence imaging to study a non-biological system such as surface adlayers. The authors use spectrally-resolved super-resolution imaging, a technique recently developed by them, with great potential to combine nanoscale optical imaging with fluorescence sensing, in this case of the local polarity of the adlayers. The power of their approach is best showcased in figure 4, in which a polarity map at the nanoscale is presented for a two-component adlayer.

While the work is well presented and has resulted in beautiful images, my main concern is that super-resolution imaging is performed in the presence of an aqueous buffer, and the effect of this buffer on the properties (e.g. morphology) of the adlayer are not investigated or even discussed. The images presented in figure 3, in which a change of morphology of the adlayers is shown, are rationalized in terms of the effect of the hydrophilic glass surface. The authors should specifically rule out the effect of the aqueous interface, for example by using surfaces of varying polarity by silanization or a similar approach.

Related to the above, the AFM images presented have been acquired in air. Perhaps images of the same sample area measured in air and in liquid could also be used to rule out the effect of the buffer.

In conclusion, the paper is currently a very nice proof-of-principle application of spectrally-resolved super-resolution imaging. However the potential effect of the imaging buffer needs to be ruled out to take it to the next level, i.e. providing real insight into surface science.

Response to reviewers' comments:

Reviewer #1 (Remarks to the Author):

1.) The localization position shown in the figure 2b can originate from the localization precision of a single molecule and not necessarily has to be limited by the confinement of the droplet.

A more detailed explanation by the authors would be required. In particular because there AFM data on small droplets (suppl. 1) is not convincing (tip artefacts).

Response: Indeed, for the smallest droplets, the apparent size we measured was limited by the localization precision of single molecules. In our experiments, we typically detected ~900 photons for each molecule in the image channel (see intensity distribution in new Fig. 1e). Previous STORM and PAINT experiments by us and others have found that a photon count at this level corresponds to a single-molecule localization precision of ~30 nm in FWHM (Refs 27 and 33). Thus, the smallest droplet size we found of ~33 nm in FWHM is certainly limited by the localization precision. Mathematically, a FWHM of 33 nm equals to a convolution of a 30 nm effective image resolution and a 14 nm underlying droplet size, implying that the actual size of the droplet could be on the order of 14 nm. However, this number is highly sensitive to small uncertainties in the exact localization precision. We agree that due to tip-induced artefacts, AFM cannot correctly map the actual sizes either. Nevertheless, AFM appears to be the only alternative method that could come close in providing nanoscale structural information for comparison with our PAINT results, and indeed the AFM results have been consistent with our PAINT results. In our revised manuscript, we have added the above discussion (Page 7, line 7-12) and pointed out that the smallest droplets are likely <~20 nm.

2.) The authors conclusion in the figure 2b, assuming the size of the smallest being limited by the localization precision, may have a different reason. The localized fluorescent signal may also be a response function of an individual molecule, which is immobilized/diffusing in a larger/smaller nanodroplet. Can the number of molecules in a nano-droplet be estimated? The correlative AFM does not prove the suggestion. A single molecule intensity analysis may give additional information on this issue. Did the authors observed diffusion of the molecules in the adlayers? All the mentioned information will be required to prove the statement. Can the authors improve their argumentation, or add new, more detailed experimental data?

Response: In our experiments, we controlled the Nile Red concentration in the imaging buffer to ensure that only one single fluorescent molecule showed up within a diffraction-limited area (~300 nm x 300 nm) at any given time, as we typically do in STORM and PAINT experiments (see also our review chapter, DOI:10.1007/4243_2013_61). Consequently, for a sub-diffraction-limit nanodroplet, we typically detected a few hundred molecules over ~30,000 frames. We further note that for each molecule, the burst of fluorescence usually only lasted one single camera frame. Due to the low quantum yield of Nile Red in water and the TIR illumination, fluorescence only occurred when single Nile Red molecules inserted from the water phase into the organic adlayers, and this fluorescence turned off again when the molecules photobleached or went back to the water phase, which happened faster than the frame rate (9 ms). Within such short fluorescence periods, no significant diffusion of single molecules occurred. Consequently, we observed single-molecule fluorescence as well-shaped airy disks with minimal motion blur (Fig. 1bc; we now removed the crosses previously drawn on the single-molecule images so the shapes are clearer). We have characterized the intensity distribution for the single molecules we detected (new Fig. 1e), and found an asymmetric single-peak distribution similar to that observed in previous STORM, PAINT, and other single-molecule experiments (Ref. 27, 31 and 32). We have added the above discussion and data to text (Page 3, line 16-19; Page 5, line 1-4; Page 20, line 8-10; Fig. 1e).

3.) Most of the droplets and structures shown in the manuscript have a substructure sizes > than the optical diffraction limit. What is the advantage of PAINT in this particular case?

Response: Thanks for this helpful discussion. We realize that we have not given due emphasis to the resolving power of PAINT in our results. Many of the structures we resolved in this work were actually characterized by dimensions significantly smaller than the optical diffraction limit (~300 nm). Fig. 2ab shows TCE adlayers as nanodroplets 30-100 nm in size. Fig. 2d shows chloroform adlayers as nano-lines ~100 nm in width, and intensity profiles gave widths of 107 and 72 nm. All these values are considerably smaller than the diffraction limit of ~300 nm. In Fig. 3, the THT, EtOAc, and DCM adlayers are all ~<100 nm in one or two dimensions. Fig. 4 further demonstrates the power of SR-PAINT: we not only visualized nanodroplets that were 30-400 nm in size and just ~100 nm apart from each other, but also resolved their spectra. Moreover, the capability to measure the precise droplet size at the nanoscale allowed us to establish a composition-size relationship (Fig. 4d) to show that droplets >~100 nm in diameter were mostly pure TCE, but the smaller droplets had higher fractions of chloroform. We have improved our discussion throughout the text (Page 7, line 5-6, line 16-18; Page 13, line 6-8, line 21-22) to emphasize the dimensions of the structures we resolved.

Minor remarks:

4.) The AFM images showing the smallest droplets are distorted by the AFM-scan. Did authors try to use different cantilevers, having tips made of various materials?

Response: We thank the reviewer for raising this discussion. We should have emphasized that in our AFM experiments we indeed considered this issue and used “soft tapping mode” to minimize disturbance to the adlayers. The force constant of our AFM probe was 5 N/m, an order of magnitude lower than typical probes used in standard tapping-mode AFM [~42 N/m; see our previous work (*Science* 329, 1188)]. In response to the reviewer’s question, we have compared results of tapping mode with a standard probe (nominal force constant: 48 N/m), as well as results of contact mode. Tapping mode with the stiffer “standard” probe gave reduced adlayer contrast, so that for chloroform, the nano-line structures were not visible at intermediate magnifications (new Supplementary Fig. 5a), and were barely visible at higher magnifications (Supplementary Fig. 5bc). In contact mode, the AFM tip dragged the adlayers along during imaging, and so the adlayers could not be correctly imaged. Together, our results are consistent with the general consensus that adlayers of small molecules are extremely difficult to characterize with AFM. Indeed, even for well-adhered larger molecules and with special instrumentation, noticeable distortion to adlayer geometry is often observed (Ref 8). We have added discussion on the comparison of AFM probes (Page 9, line 9-11), with new data shown as Supplementary Fig. 5.

5.) What is the SNR for the fluorescent Nile Red signals? What illumination time? How much time is required for full sample characterization? How much faster is the characterization of a single diffraction limited image?

Response: We collected ~900 photons from each molecule in the image channel (distribution in new Fig. 1e). For each molecule, the light intensity is distributed as a diffraction-limited airy disk over ~4x4 pixels, with the brightest central pixel receiving a signal of ~150 photons. The typical noise (variation in the background) is σ ~6 photons per pixel. SNR is thus estimated as ~25. Alternatively, the four central pixels receive signals of >300 photons, and σ for the sum of four pixels is ~12 photons, so the SNR is >25. The sample was continuously illuminated, and the EMCCD recorded in the wide-field (512x256 pixels for the entire frame; divided as two halves of 256x256 pixels for single-molecule images and spectra, respectively) continuously at 9 ms/frame (110 frames per second). As discussed above, most single-molecule fluorescence lasted only one frame (<9 ms). We typically recorded 20,000–40,000 frames of raw, diffraction-limited single-molecule images to generate a PAINT image, corresponding to a 3-6 min recording time. We have added these experimental details to Methods (Page 20, line 4-5, 14-19).

6.) What are the time points of measuring? Are the observed structure stable in time?

Response: Data collection time is 3-6 min for each image (discussed above). The adlayers are stable for hours during the experiments. These new results and discussions have been added to the main text (Page 8, line 1-9; Page 20, line 16-17) and Supplementary Fig. 2.

7.) Axis labeling in figures is not consistent (e.g. brackets in Fig. 2 g and h, 3f and 4c in comparison to the physically correct division by the unit as shown in Fig. 1d)

Response: Thanks for pointing out this issue. We have fixed it, and we now use brackets for all figures.

8.) Color scale bars are missing regarding the height in AFM images.

Response: We have added color scale bars to AFM images for which height profiles are not provided (Supplementary Fig. 4 and 5). We do not often add color scale bars to AFM images for which height profiles are given, since the latter is a more quantitative presentation.

Reviewer #2 (Remarks to the Author):

Point-wise Comments:

o This paper deals with study of structure and composition of adlayers of small molecules of differing polarity on glass surface using the novel spectrally resolved super-resolution microscopy technique that the authors have reported earlier [Zhang, Z. Y., Kenny, S. J., Hauser, M., Li, W. & Xu, K. Ultrahigh-throughput single-molecule spectroscopy and spectrally resolved super-resolution microscopy. Nat. Methods 12, 935-938 (2015)].

o Authors have established the dimensionality and polarity of the different adlayer molecules on the glass surface by imaging and collecting the emission spectra of polarity-sensitive solvatochromic fluorophore Nile Red molecule stochastically inserted into adlayers of various small organic molecules by employing spectrally resolved points accumulation for imaging in nanoscale topography (SR-PAINT) technique along with tapping mode-AFM for height determination of the adlayers.

o They also report interesting and direct visualization of the demixing at nanoscale for adlayers formed from a solution of two miscible solvents.

o The mapping of the changing structural details and their correlation with the polarity of the different organic molecules by a non-invasive optical SR-PAINT technique and its confirmation with AFM topography is novel concept.

o This technique may provide very valuable information in the field of self-assembled monolayers and their applications.

o In the introduction when referring to Scanning probe techniques having been used to probe adlayer structure, a reference to the role of ambient water adlayers formed on graphene interfaces on hydrophilic and hydrophobic substrates as reported by Gargi Raina et al. [Phys.Chem.Chem.Phys., 2015, 17, 13964] is appropriate.

o The ability to reproduce the work by other researchers is fair provided the experimental set-up can be assembled appropriately.

Response: We thank the reviewer for his/her enthusiastic comments and excellent summary of our results. We have added the suggested reference as another example of work based on scanning probe techniques (ref. 10).

Points of Concern:

o Though this optical technique would help in visualizing the structure of small molecule adlayers on various technologically important substrates non-invasively, the role of the chosen fluorophore molecule's effect on the image generation process needs to be understood more clearly.

Response: This is a good suggestion. In this work, we have focused results on Nile Red as it provides strong spectral shifts to report local polarity. Prompted by the reviewer's question, we have successfully demonstrated that another dye, Merocyanine 540, works equally well for the PAINT imaging of adlayers (new Supplementary Fig. 3). Similar structural features are observed, thus indicating that the adlayer structures visualized by PAINT are independent of the particular dyes used. However, although Merocyanine 540 is also characterized by a substantial increase in fluorescence in the organic phases when compared to the water phase, it exhibits minimal shifts in emission spectrum for solvents of different polarities [*Chem. Phys.* 2001, 263, 415]. Consequently, it is not useful for SR-PAINT. We have added these new results and discussions in the main text (Page 8, line 10-16) and Supplementary Fig. 3.

o The demixing at nanoscale for adlayers formed from a solution of one particular ratio of 1:3 of TCE and Chloroform miscible liquids shows the formation of nanodroplets of different sizes and varying shapes. It would be interesting to have a detailed study of various compositions of the two miscible liquids and their effect on the size and arrangement of adlayer molecules on the surface.

Response: This is a good suggestion. We have performed new experiments on the influence of mixture composition on adlayer morphology and composition (new Supplementary Fig. 13). For a 1:1 TCE-chloroform mixture, we observed fewer high-polarity droplets when compared to the 1:3 TCE-chloroform mixture, and small, intermediate-polarity droplets surrounded large low-polarity droplets as opposed to forming fragmented networks. This result is consistent with the lower concentration of chloroform in the starting mixture: at room temperature the vapor pressure of chloroform is 3-fold higher than TCE, so it evaporates away faster during adlayer formation. For a 1:6 TCE-chloroform mixture, low-polarity large droplets disappeared, and segments of nano-lines interspersed with small droplets, consistent with the high chloroform content of the starting mixture. Together, our SR-PAINT results on adlayers of different starting solutions show trends consistent with the expected physical properties of the two solvents, but further reveal fascinating evolution of nanoscale structures and composition distributions unavailable to other methods. We have included these remarkable new results and discussions in the main text (Page 14, line 13-17) and Supplementary Fig. 13.

In conclusion, this manuscript can be accepted as it is on account of the demonstrated novel application of SR-SRM tool for the study of the structure of the adlayer composition of small molecules on surfaces and thus providing useful and important insights into the surface structure dynamics of the different types of adlayers in general, at the nanoscale.

Response: We thank the reviewer's very enthusiastic recommendation.

Reviewer #3 (Remarks to the Author):

This paper reports an original application of super-resolution fluorescence imaging to study a non-biological system such as surface adlayers. The authors use spectrally-resolved super-resolution imaging, a technique recently developed by them, with great potential to combine nanoscale optical imaging with fluorescence sensing, in this case of the local polarity of the adlayers. The power of their approach is best showcased in figure 4, in which a polarity map at the nanoscale is presented for a two-component adlayer.

While the work is well presented and has resulted in beautiful images, my main concern is that super-resolution imaging is performed in the presence of an aqueous buffer, and the effect of this buffer on the properties (e.g. morphology) of the adlayer are not investigated or even discussed. The images presented in figure 3, in which a change of morphology of the adlayers is shown, are rationalized in terms of the

effect of the hydrophilic glass surface. The authors should specifically rule out the effect of the aqueous interface, for example by using surfaces of varying polarity by silanization or a similar approach.

Related to the above, the AFM images presented have been acquired in air. Perhaps images of the same sample area measured in air and in liquid could also be used to rule out the effect of the buffer.

In conclusion, the paper is currently a very nice proof-of-principle application of spectrally-resolved super-resolution imaging. However the potential effect of the imaging buffer needs to be ruled out to take it to the next level, i.e. providing real insight into surface science.

Response: We thank the reviewer for his/her enthusiastic comments. Our confidence in accurately reporting the structure of the adlayers partly arose from the observation that our PAINT results are always consistent with that of AFM, in which samples were typically examined in the air immediately after preparation without seeing an aqueous phase. Prompted by the reviewer's comments, we have now characterized the same sample area before and after being immersed in the imaging buffer. As we have shown that AFM does not faithfully visualize adlayers and strongly disrupts the sample, we turned to alternative optical methods. Among the different adlayers investigated in this study, we found that TCE nanodroplets could be observed by differential interference contrast (DIC) microscopy, albeit at much lower resolution (new Supplementary Fig. 2). Individual nanodroplets had similar apparent sizes of ~600 nm, limited by the resolution of DIC microscopy, but showed different contrasts presumably due to differences in their actual sizes. Adding the Nile Red imaging buffer did not alter the adlayer structure (new Supplementary Fig. 2). Moreover, through SR-PAINT we have confirmed that the adlayer nanostructures of different solvents to be highly stable in the imaging buffer over hours (new Supplementary Fig. 2). The observed high structural stability is expected: the very small contact angles of adlayers, as confirmed in this work and also observed in previous studies, suggest strong adlayer-substrate interactions that stabilized the adlayer structure. We have added the above new results and discussion into the revised manuscript (Page 8, line 1-9). We agree that investigating solvent adsorption on a silanized hydrophobic surface would also be interesting. However, our preliminary results indicated that silanized surfaces result in strong fluorescence backgrounds; we plan to address this issue in future experiments. We further note that in this revision we have further added new data on TCE-chloroform mixtures of different ratios (new Supplementary Fig. 13); these results revealed fascinating nanoscale structures and compositions consistent with the expected physical properties of the two solvents, thus further demonstrating the strength and consistency of our method.

Reviewers' comments:

Reviewer #1 (Remarks to the Author):

Major:

- 1.) Localization microscopy in the current case relies strongly on buffer conditions; thus the experiment have to be performed under wet conditions. That means already formed adlayers have to be covered with an additional buffer (→ wet). What is the adlayer? Are the spectral positions/shifts determined by the authors introduced by the (wetted) adlayer or by the used buffers, which slowly dissolve into the adlayer? (authors claim –‘have confirmed that the adlayer nanostructures of different solvents to be highly stable in the imaging buffer over hours’)Discuss the limitations.
- 2.) In the revision, the authors only provide an assumption of the localisation precision (~30nm); since the required numbers are available, please calculate it using Thompson/Mortenson formula.
- 3.) The discussion on the influence of the diffusion of single molecules – commented with ‘no significant diffusion of single molecules occurred’ – is not precise. One has to calculate it and then verify the result.
- 4.) Typically to correct the blurring in AFM images a deconvolution of the image with the tip geometry is performed. Did the authors try to correct for the blurring using a deconvolution algorithm? Did you perform the AFM-imaging only in a ‘dry-state’ before and after the optical readout or did you perform AFM-scans also in the imaging buffer?

Minor:

- 5.) I suggest adding some profiles/cross-sections in figure 4 – indicating the sub-100nm resolution of this picture.
- 6.) The intensity (number of counts) in the insert figure 2 b is missing.

Reviewer #2 (Remarks to the Author):

The points of concerns raised in review of the paper concerning the choice of the fluorophore molecules' effect on the image generation have been addressed to and the necessary modification included in the text as well as by including a supplementary figure for the same.

The authors have also performed new experiments on the influence of mixture composition on adlayer morphology and composition and accordingly added an additional figure in the supplementary figures.

Reviewer #3 (Remarks to the Author):

The authors have addressed my previous comments in a satisfactory way, and I congratulate them for their work.

As an additional suggestion, they may want to consider referring to a recent review on the use of super-resolution imaging in non-biological applications: *Small Methods* 2017, 1(10) 1700191.

Reviewer #1 (Remarks to the Author):

Major:

1.) Localization microscopy in the current case relies strongly on buffer conditions; thus the experiment have to be performed under wet conditions. That means already formed adlayers have to be covered with an additional buffer (wet). What is the influence of the buffer on the formed adlayer? Are the spectral positions/shifts determined by the authors introduced by the (wetted) adlayer or by the used buffers, which slowly dissolve into the adlayer? (authors claim –‘have confirmed that the adlayer nanostructures of different solvents to be highly stable in the imaging buffer over hours’)Discuss the limitations.

Response: As discussed in our previous response to Reviewer #3, our results indicated that the imaging buffer did not appreciably affect adlayer properties. To begin with, all our PAINT results were consistent with AFM results obtained in the dry state. With Supplementary Fig. 2, we further showed through DIC microscopy that adding the imaging buffer did not alter the adlayer structure, and then through SR-PAINT that the nanoscale structure and polarity of the adlayers were highly stable in the imaging buffer over hours. Prompted by the reviewer’s discussion, we have now further examined the averaged spectra at the beginning and after 1h of adding the imaging buffer (new Suppl. Fig. 2eh), to see if small changes to polarity occurred (i.e., the possibility of “slowly dissolve into the adlayer”). No appreciable differences were observed. These results, together with the fact that we observed highly consistent and reasonable structural and polarity trends for solvents of different polarity and for solvent mixtures of different compositions, indicated that the imaging buffer did not significantly affect our results. We have added the averaged spectra as Suppl. Fig. 2eh, and further emphasized in text the limitation that our experiments were based on the use of an imaging buffer. “We note that the adlayers were stable with the application of the imaging buffer necessary for our SR-PAINT approach.” “we further confirmed the adlayer nanostructures of both TCE and chloroform to be stable in the imaging buffer over hours, in terms of both nanoscale structure and polarity (Supplementary Fig. 2)”.

2.) In the revision, the authors only provide an assumption of the localisation precision (~30nm); since the required numbers are available, please calculate it using Thompson/Mortenson formula.

Response: The reviewer made a good point that the localization precision may be theoretically calculated from the number of photons collected. Indeed, in the literature, researchers sometimes use such values to characterize resolution. However, as we discussed this very issue in a review chapter [Xu, Shim, Zhuang, in *Far-Field Optical Nanoscopy* (Springer, Berlin, 2015), pp. 27-64, DOI:10.1007/4243_2013_61], the theoretical equations generally give unrealistic values that are not experimentally achieved, so we avoid using these values when interpreting data. Specifically, in this case (~620 nm emission at 900 photon counts), the Thompson/Mortenson formula gives FWHM of ~17 nm. Previous results by us and others, however, have found that this value cannot be reached experimentally even with much higher photon counts. Instead, a ~30 nm experimental FWHM localization precision was found (Refs. 28 and 37 in the revised manuscript). As noted in our review chapter, factors like pixel nonuniformity and mechanical instability contribute to the final result, so the theoretical values are only meaningful as a lower bound. We have revised the discussion and added references to theory: “While the ~900 photons we detected per molecule (Fig. 1e) could theoretically translate to a single-molecule localization precision of ~17 nm in FWHM^{35,36}, experimentally a ~30 nm FWHM localization precision has been found under similar settings^{28,37}.”

3.) The discussion on the influence of the diffusion of single molecules – commented with ‘no significant diffusion of single molecules occurred’ – is not precise. One has to calculate it and then verify the result.

Response: We note that as the organic adlayers in this work were extremely thin (a few nm), diffusion mainly occurred vertically between the aqueous imaging buffer and the organic adlayers, as opposed to laterally within the organic adlayers. Consequently, fluorescence switched on suddenly when single Nile Red molecules diffuse into the organic adlayer phase, but then immediately switched off as the molecule diffused back into the aqueous buffer. It is very difficult to estimate how fast single molecules diffuse in the vertical direction. However, if we assume comparable diffusion rates in vertical and lateral directions, then a molecule will only have the time to diffuse a few nm laterally as it vertically diffuses through the few-nm height. We have improved this discussion in text. “Due to the extreme thinness of the adlayers (a few nanometers; below), single molecules rapidly diffused in

(strongly fluorescent) and out (non-fluorescent) of the adlayer. This short timeframe left little chance for lateral diffusion. Consequently, single-molecule fluorescence appeared as diffraction-limited spots with minimal motion blur (Figs. 1bc)."

4.) Typically to correct the blurring in AFM images a deconvolution of the image with the tip geometry is performed. Did the authors try to correct for the blurring using a deconvolution algorithm? Did you perform the AFM-imaging only in a 'dry-state' before and after the optical readout or did you perform AFM-scans also in the imaging buffer?

Response: The AFM tips we used had radii of <10 nm, so we do not expect the tip geometry to contribute significantly to our images at intermediate magnifications for features >~100 nm in dimension. Instead, the notable distortions there were consistent with tip-sample interactions, including dragging along the scanning direction (Fig. 2c) and removal of structures (Fig. 2ef). For images at high magnifications (e.g., Suppl. Fig. 4bc), although the true sizes of the adlayers could now be comparable to the tip size, severe distortions still appeared to be dominated by tip-dragging along the scanning direction. Overall, the noticeable distortions cannot be easily fixed by deconvolution algorithms, and our results are consistent with the general consensus that adlayers of small molecules are extremely difficult to characterize with AFM. Our AFM experiments were performed in a dry-state in air; this is the standard approach that is available to most researchers and provides the best image quality. We have added above information in the revised manuscript. "AFM images were taken in the dry state in air" "No attempts were made to deconvolve the tip geometry in the presented data."

Minor:

5.) I suggest adding some profiles/cross-sections in figure 4 – indicating the sub-100nm resolution of this picture.

Response: This is a good suggestion. We have added, as an inset of Fig. 4b, an intensity profile for four closely located nanodroplets in the image. This helped show that each individual nanodroplet is ~50 nm FWHM in size, and that they are well resolved from each other at center-to-center distances of ~100 nm. This is mentioned in the text: "Both the sizes of the droplets and the separations between the droplets were often substantially smaller than the diffraction-limited resolution of conventional light microscopy (Fig. 4b inset)."

6.) The intensity (number of counts) in the insert figure 2 b is missing.

Response: We have added this.

Reviewer #2 (Remarks to the Author):

The points of concerns raised in review of the paper concerning the choice of the fluorophore molecules' effect on the image generation have been addressed to and the necessary modification included in the text as well as by including a supplementary figure for the same.

The authors have also performed new experiments on the influence of mixture composition on adlayer morphology and composition and accordingly added an additional figure in the supplementary figures.

In conclusion, I am satisfied with the response provided by the authors and the corresponding new experiments performed to add the relevant information in the manuscript. Hence, I recommend the publication of this manuscript.

Response: We thank the reviewer for his/her enthusiastic comments.

Reviewer #3 (Remarks to the Author):

The authors have addressed my previous comments in a satisfactory way, and I congratulate them for their work.

As an additional suggestion, they may want to consider referring to a recent review on the use of super-resolution imaging in non-biological applications: Small Methods 2017, 1(10) 1700191.

Response: We thank the reviewer for his/her enthusiastic comments. We have added this new reference (#23).

REVIEWERS' COMMENTS:

Reviewer #1 (Remarks to the Author):

Minor Changes:

1.)

, could theoretically translate to a single-molecule localization precision of ~17 nm in FWHM^{35,36}, experimentally a ~30 nm FWHM localization precision has been'

The added text is ok, however the change shall include a more precise info regarding the inaccuracy of the localization. As it is typically done in statistics.

2.) Authors write "AFM images were taken in the dry state in air" "No attempts were made to deconvolve the tip geometry in the presented data."

I believe it is necessary to conclude that it cannot be excluded that due to the different measurement conditions (wet for FM and dry for AFM) changes of the sample may occur. Since, the authors draw conclusions based on the AFM data for the FM data.

REVIEWERS' COMMENTS:

Reviewer #1 (Remarks to the Author):

Minor Changes:

1.) could theoretically translate to a single-molecule localization precision of ~ 17 nm in FWHM^{35,36}, experimentally a ~ 30 nm FWHM localization precision has been'

The added text is ok, however the change shall include a more precise info regarding the inaccuracy of the localization. As it is typically done in statistics.

Response: We have added discussions regarding the possible reasons that cause the difference between the theoretical and experimental localization precisions, which we previously discussed in the reply letter but did not add to the main text. Page 5, Line 4: “experimentally a ~ 30 nm FWHM localization precision has been found under similar settings^{28,37} due to imperfections of the imaging system including pixel nonuniformity and mechanical instability³⁸.”

2.) Authors write “AFM images were taken in the dry state in air” “No attempts were made to deconvolve the tip geometry in the presented data.”

I believe it is necessary to conclude that it cannot be excluded that due to the different measurement conditions (wet for FM and dry for AFM) changes of the sample may occur. Since, the authors draw conclusions based on the AFM data for the FM data.

Response: As discussed previously, all our PAINT results were consistent with AFM results obtained in the dry state, although in some cases the tip-sample interactions in AFM prevented us to compare structural details at the nanoscale. We have revised the main text to further emphasize that the AFM was performed in the dry state, but the results were consistent. Page 6, Line 12: “We next compared results with AFM acquired in the dry state in air.” Page 8, Lines 13-15: “Overall, our AFM results, acquired in the dry state in air, were generally consistent with our PAINT SRM results, although tip-sample interactions often obscured nanoscale structural features visualized by PAINT.”